**Data Availability Statement:** Partial Sex Now data is publicly available on the Our Stats dashboard (https://ourstats.ca/our-dashboard). The University

# Health and well-being of trans and non-binary participants in a community-based survey of gay, bisexual, and queer men, and non-binary and Two-Spirit people across Canada

Leo Rutherford[1], Aeron Stark[1], Aidan Ablona[2,3], Benjamin J. Klassen[3], Robert Higgins[1], Hanna Jacobsen[1,3], Christopher J. Draenos[3], Kiffer G. Card[1,3], Nathan J. Lachowsky[1,3]*

**1** School of Public Health & Social Policy, University of Victoria, Victoria, British Columbia, Canada, **2** British Columbia Centre for Disease Control, Vancouver, British Columbia, Canada, **3** Community-Based Research Centre, Vancouver, British Columbia, Canada

* nlachowsky@uvic.ca

## Abstract

There is a paucity of population health data on the experiences of transgender, non-binary, and other gender minority gay, bisexual, and queer men, and Two-Spirit people in Canada. To address this gap, this article presents a socio-demographic and health profile of trans and non-binary participants from the community-based bilingual 2018 Sex Now Survey. Participants were recruited in-person from Pride festivals in 15 communities to self-complete an anonymous paper-and-pen questionnaire. To be eligible, participants needed to be at least 15 years old, live in Canada, either report a non-heterosexual sexual identity or report sex with a man in the past 5 years, and not report gender identity as a woman. Through community consultations the survey was inclusive of trans men, non-binary people, and Two-Spirit people. Three gender groups (cisgender, transgender, and non-binary) were created, and trans and non-binary participants were compared with their cisgender peers across a variety of demographic, mental health, sexual health, and general health indicators. Odds ratios were calculated to determine initial significance for categorical variables, and adjusted odds ratios were calculated to control for five possible confounders (age, ethnoracial identity, country of birth, sexual identity, and financial strain). Significant differences emerged across all sets of indicators, with many of these findings remaining significant after adjusting for confounding variables, including significantly higher reported rates of mental health challenges and sexual health service barriers for trans and non-binary participants compared to the cisgender group. Trans and non-binary participants were also more likely to be in polyamorous relationships. Collectively, our findings demonstrate that trans and non-binary people experience significant disadvantages compared with cisgender sexual minority men. Improved educational supports and employment protections, access to queer and gender affirming healthcare, and trauma-informed mental health services are needed to improve the health wellbeing of trans and non-binary people in Canada.

of Victoria's Human Research Ethics Board has only approved storage of our data on secure university servers since the data contain potentially sensitive information about study participants. Data is available on request through secure university servers only. Any requests to access the data can be made to University of Victoria's Human Research Ethics Board (250-472-4545 or ethics@uvic.ca).

**Funding:** Sex Now 2018 received funding support from Canadian Blood Services MSM Research Fund, funded by the federal government (Health Canada) and the provincial and territorial ministries of health. The views herein do not necessarily reflect the views of Canadian Blood Services or the federal, provincial, or territorial governments of Canada. Additional in-kind contributions were received from the Public Health Agency of Canada's National Microbiology Laboratory. Additional funding for this analysis was provided by Women and Gender Equality (WAGE) Canada. NJL is supported by a Scholar Award from the Michael Smith Foundation for Health Research (#16863).

**Competing interests:** The authors have declared that no competing interests exist.

## Introduction

A robust body of evidence demonstrates that gay, bisexual, queer and other sexual minority men in Canada face a greater burden of health inequities–for example, in relation to mental health and HIV transmission–when compared with their heterosexual peers. However, there is a paucity of data on the health experiences of trans, non-binary, and other gender minority people in Canada. Indeed, very little population health research has been conducted with trans, non-binary, and Two-Spirit people in Canada. Notable exceptions include the 2010 Trans PULSE study of trans people in Ontario, which included both quantitative and qualitative data collection [1–3], the Canadian Trans and Non-binary Youth Health Survey [4], and the recent national Trans PULSE Survey, for which data analysis and reporting is ongoing [5].

Existing evidence indicates that there are significant socioeconomic and healthcare disparities between trans and non-binary people and their cisgender peers, such as barriers to accessing healthcare. For example, findings from Sex Now 2015, a national health and wellbeing survey of gay, bisexual, and queer men and non-binary and Two-Spirit people (GBT2Q) in Canada, indicate that trans GBT2Q participants were significantly more likely to report experiencing healthcare discrimination in their lifetime and in the past 12 months when compared with cisgender GBT2Q participants [6]. While not solely focused on GBT2Q men, the national and Ontario (Canada's most populous province) Trans PULSE studies have found similar healthcare access challenges. The Trans Pulse Canada Team found that 45% of respondents reported one or more unmet healthcare needs in the past year and 12% had avoided the emergency room in the past year [5]. The Ontario-based Trans PULSE study found negative experiences with healthcare providers among trans individuals, as 52% of respondents who presented in their felt gender reported having experienced transphobia in the emergency room [1]. Bauer et al. estimate that 38% of trans people in Ontario had prior trans-specific negative experiences with family physicians, and approximately half were uncomfortable discussing trans issues with their doctor [2]. These statistics highlight that barriers to acceptable healthcare persist for many trans and non-binary people.

Additionally, trans and non-binary people often report healthcare disparities, and poorer physical and mental health than their cisgender peers [5, 6]. Trans PULSE Canada respondents tended to rate their general health better than their mental health, as 73% of respondents indicated their health was good, very good, or excellent, but only 44% reported the same about their mental health [5]. Indeed, 31% had considered suicide and 6% had attempted suicide in the past year. The 2019 Canadian Trans and Non-binary Youth Health Survey found even more concerning levels of poor mental health, with only 16% of respondents reporting their mental health as good or excellent [4]. Most youth (88%) reported having a chronic mental health condition, such as depression or anxiety. Within the past year alone, 64% of youth had considered suicide and 21% had attempted suicide [4]. Collectively, these data indicate that there are heightened mental health challenges for young trans people compared with the general trans population.

Regarding sexual health, existing Canadian data indicates that trans men are less likely to report condomless sex and sex with casual partners, which suggests lower prevalence of sexual encounters that might lead to HIV transmission [6, 7]. Trans PULSE found that only 7% of GBT2Q transmasculine participants had had a high-risk sexual experience in the past year, and 25% had not had a sex partner at all in the past year [7]. All HIV-related risk was associated with the subgroup of participants who had cisgender men as sex partners, but only one-third of participants fell into this category, making the remaining two-thirds categorically low-risk. While these behaviour profiles suggest a general low HIV risk, existing evidence also indicates that trans men test less frequently for HIV and STIs, while also highlighting several

barriers to testing for trans men [6–8]. Qualitative interviews by Rich et al. with gay, bisexual and queer transgender men in Vancouver, BC illuminated several possible explanations for these findings. Some of these Vancouver participants were living in an urban centre and well-connected to healthcare systems, such as routine transition-related care appointments that made integrating HIV testing into their regular healthcare regimen easy [9]. However, other participants also described barriers to HIV testing and accessing healthcare in general, such as a lack of trans competency in care providers and past experiences of healthcare discrimination. Both Scheim et al. and Rich et al. argue that trans men are often excluded from existing health services and campaigns, and trans-competent sexual healthcare should be integrated into services targeted at both the general population and at sexual minority men [7, 9].

Experiences of discrimination, marginalization, and oppression for trans and non-binary people are also observable in socioeconomic indicators [10]. Despite being a highly educated population, with 89% of Trans PULSE Canada respondents aged 25 and older having at least some post-secondary education, trans and non-binary people face high rates of poverty and homelessness. The National Trans PULSE survey found that half of respondents 25 years of age or older had a personal income of less than $30,000/year, and 40% lived in a low-income household [5]. Furthermore, 10% of respondents were currently housing insecure (including homeless or living in temporary and unstable housing) and 15% responded their household sometimes or often did not have enough to eat in the last 12 months [5].

Despite the many manifestations of oppression and marginalization in trans people's lives, research suggests that some of these impacts may be mitigated by positive factors such as trans-affirming social support and connection. Veale et al.'s research on trans and non-binary youth between the ages of 14 and 25 in Canada found that enacted stigma, as measured through experiences of harassment, discrimination, bullying, and physical and sexual violence, was associated with higher likelihood of negative mental health outcomes, such as non-suicidal self-injury, recent suicide attempt, and extreme stress and despair [11]. However, supportive environments and social connections with family, friends, and school reduced the negative effects of enacted stigma on trans youth's mental health [11]. Similarly, the Trans PULSE Ontario project found that several structural and interpersonal factors had a strong impact on suicide ideation or attempts for trans individuals. Their modelling found that factors such as fewer experiences of transphobic discrimination and violence, strong parental support for gender identity or expression, having an identity document with a gender marker that matched one's lived gender, and having completed medical transition when it was desired were all associated with a reduced likelihood of suicidal ideation and attempts [12].

The limited data available suggests that trans and non-binary people in Canada experience discrimination, oppression and marginalization that results in barriers in accessing healthcare, experiences of verbal, physical, and sexual violence, poverty, physical and mental illness, and a variety of other negative socioeconomic and health indicators. To further substantiate this area of inquiry, this article presents a socio-demographic and health profile of trans and non-binary participants from a community-based population health data source, the 2018 Sex Now Survey, which provides valuable information for guiding interventions to address inequities among trans and non-binary communities.

## Methods

Sex Now is a national periodic cross-sectional survey conducted by the Community-Based Research Centre (CBRC) to promote the health and well-being of GBT2Q people. It is Canada's largest and longest running survey of GBT2Q health and well-being, providing an essential source of data that is widely used by community, public health, research, and policy

stakeholders. The survey was originally commissioned in 2002 by the BC Centre for Disease Control (CDC) as an investigation into rising HIV infection rates among gay men in the province of British Columbia. Since then, there have been several survey cycles starting with BC Pride Festivals in 2002 & 2004, moving to the internet in 2006, 2007 & 2008, and then to cover all of Canada in 2010, 2012, 2015, and 2018.

For Sex Now 2018, we consulted with community organizations and stakeholders across the country to develop the survey. This included targeted consultation with trans men, non-binary people, and Indigenous Two-Spirit community members. Subsequently, many of these community organizations led in-person recruitment at Pride festivals and related events across 15 Canadian cities between May and September 2018. Recruitment cities (from west to east) included Vancouver, New Westminster, Surrey, Abbotsford, Kamloops, Kelowna, Calgary, Edmonton, Winnipeg, London, Toronto, Ottawa, Cornwall, Montreal, and Halifax. Community partners provided staff and volunteers for specific sites and oversaw the administration of the self-completed anonymous paper-and-pen surveys. Community partners also promoted the survey in advance of these events through their social media and listservs; additional online promotion was also done by CBRC. Past versions of the survey have been instrumental in supporting GBT2Q health groups to better support their programs, secure funding, and advocate for policy change. The current questionnaire is freely available on the CBRC website, and includes sections on demographics, sex life, sexual health, blood donation, HIV and Hepatitis C, mental health, substance use, social health, healthcare, discrimination, and violence. Participants were asked four questions about how often they had been bothered by different symptoms of depression and anxiety over the last two weeks. The first two questions ("Little interest or pleasure in doing things" and "Feeling down, depressed, or hopeless") were taken from the Patient Health Questionnaire-2 (PHQ-2) [13] to assess for depressive symptoms. Questions three and four ("Feeling nervous, anxious or on edge" and "Not being able to stop or control worrying") were taken from the Generalized Anxiety Disorder-2 (GAD-2) [14] to assess symptoms of anxiety. Response options were "not at all," "several days," "more than half the days," and "nearly every day." From these questions, two mental health measures were established: one for depression and one for anxiety. For each measure, the possible range of scores was 0 through 6, with a score of 0 indicating a respondent answered "not at all" to both questions, and a score of 6 meaning the person answered "nearly every day" to both questions. A score of 3 could mean that a participant had either experienced one symptom "nearly every day" or selected "more than half the days" as one response and "several days" as the other response. A cut-off of 3 or higher is indicative of a possible depression or anxiety disorder, and was chosen by the developers of these measures to determine if further evaluation is necessary. For both depression and anxiety, a binary variable was created using the cut-off value of 3 (the standard cut-off point for these measures). All questions in the survey were optional and some participants elected not to respond to some questions.

Funding for the Sex Now 2018 survey cycle was provided by Canadian Blood Services (CBS) to generate evidence on potential policy alternatives to blood donor deferral for "men who have sex with men." At the time of survey administration, trans blood donors were screened based on their sex assigned at birth, unless they had had any lower gender affirming genital surgery procedure(s), previously referred to as sex reassignment. We recognize that these risk assessment policies are cisnormative, do not affirm the identities of trans and non-binary people, and do not attend to the complexity of trans and non-binary bodies.

In order to be eligible for Sex Now 2018, participants had to: 1) self-identify as men (inclusive of people reporting trans experience), non-binary (regardless of sex assigned at birth), or Two-Spirit; 2) identify as gay, bisexual, queer, or another non-heterosexual identity and/or have reported having had sex with another man (cis or trans) in the last 5 years; 3) be 15 years

of age or older; 4) be living in Canada; 5) be able to provide informed consent and complete the questionnaire in either French or English, and; 6) must not have already participated in the study at another venue. The ethics protocol for this study was reviewed and approved by the research ethics boards at the University of Victoria (BC17-487), University of British Columbia (BC17-487), and University of Toronto (35929). Oral consent was obtained from all study participants. No parental consent was required from parents or guardians of minors, our REB approved individual assent for participants aged 15 and older. Participants were given Sex Now-branded dog tags as a as a token of appreciation for participating in the study.

This paper analyzes Sex Now 2018 socio-demographic and health and well-being data to highlight salient similarities and differences across three overarching gender history and identity groups: cisgender, transgender, and non-binary. Participants were grouped into these categories based on their responses to two survey questions: gender identity and transgender lived experience. The first question, "What is your gender identity?," had three options: "man," "woman," and "neither. I prefer to self-describe as: _____________." A participant could only select one answer as a response. As per above, participants were eligible if they answered "man" or "neither;" those who answered "neither" had an opportunity to provide a written response for how they self-describe their gender identity. The second question asked was, "Do you have trans experience? (i.e., your gender is different than the sex you were assigned at birth)." A participant could answer either "yes" or "no." In order to be included in this report's analysis, participants had to answer both of these questions. Using these two questions, we created three participant groups. The *cisgender group* is composed of all participants that selected "man" as their gender identity and responded "no" to the question about trans experience. The *transgender group* includes all participants who selected "yes" to trans experience, regardless of their gender identity being man or non-binary. We defined the *non-binary group* as those who responded "neither" man nor woman to the question of gender identity, irrespective of their trans experience. This included a variety of identities including those who wrote in responses such as 'genderqueer' or 'enby,' who were all categorized as non-binary for this study. Therefore, the trans and non-binary groups are not mutually exclusive. This means that some trans participants are also included in the non-binary group and vice versa.

The survey data were analyzed using Statistical Package for the Social Sciences (SPSS) version 26 for Mac. The cisgender group was compared with both the trans and the non-binary group. A statistical comparison of the trans group with the non-binary group was not possible due to the overlap of participants across these two groups (i.e., these groups are not mutually exclusive). However, although statistical comparisons were not possible, we include some descriptive comparisons of the two groups. Odds ratios were calculated to determine initial statistically significant differences between trans versus cisgender and non-binary versus cisgender groups. Univariate and multivariate logistic regression models were used to quantify the relationship between trans participants (versus cisgender) and non-binary participants (versus cisgender) across multiple health domains. All models were adjusted for five confounders: age, white ethnoracial identity vs. non-white ethnoracial identity, born in Canada vs. not, gay sexual identity vs. not, and financial strain. Each variable reported on in the results section was assessed in a separate model and adjusted for the same five confounders. Odds ratios and 95% confidence intervals (95%CI) are reported, with interpretation of significant relationships where 95%CIs do not include 1. As these statistical tests assume that data were collected through random probability sampling, results should be taken with caution as they may be biased by the fact that respondents are not representative of the broader population.

## Results

There was a final analytic sample of 3,083 cisgender participants, 296 trans participants, and 150 non-binary participants, with 106 participants who identified both as non-binary and as having trans experience.

### Recruitment site & demographics

City of recruitment differed between trans and cisgender participants in notable ways (Table 1). A higher percentage of trans respondents participated in the smaller cities of Halifax, London, and the interior of British Columbia (Kelowna and Kamloops) compared with cisgender participants, while Ottawa had a smaller percentage of trans respondents participate. When odds ratios were calculated, the only statistically significant difference was that more trans participants were recruited in London compared with other cities. There was no statistically significant difference in city of recruitment between non-binary and cisgender participants.

There were statistically significant differences in the age profiles of trans and non-binary participants when compared with cisgender participants (Table 1). Both trans and non-binary groups were younger than the cisgender group; the proportions of both trans and non-binary participants under the age of 25 was more than double the proportion of cisgender participants under the age of 25. Compared with cisgender participants, trans participants were more likely to be under the age of 25 (42.2% vs. 18.0%) and less likely to be aged 30–39 (19.9% vs. 28.6%), 50–59 (7.1% vs. 12.9%), and over 60 (3.0% vs. 6.5%). Among non-binary respondents, two-thirds (66.0%) were under the age of 30. Compared with cisgender participants, non-binary participants were more likely to be under the age of 25 (40.7% vs. 18.0%) and less likely to be aged 30–39 (18.7% vs. 28.6%) or 50–59 (3.3% vs. 12.9%). For both trans and non-binary groups, calculated odds ratios indicated statistically significant differences across all age categories when compared with cisgender participants.

Ethnoracial identities were not mutually exclusive as participants could identify with multiple groups (Table 1). Compared with cisgender participants (8.9%), a lower percentage of trans participants identified as East or Southeast Asian (5.1%). The percentage of trans participants identifying as Indigenous was twice that of the cisgender sample (16.6% vs. 8.1%). Compared with cisgender participants, non-binary participants were more likely to identify as African, Caribbean or Black (7.3% vs. 4.0%), Indigenous (16.0% vs. 8.1%), or "other" (2.7% vs. 0.9%).

A statistically significant difference existed between the proportion of cisgender and trans participants who were born in Canada (Table 1), with trans participants being more likely to be born in Canada than cisgender participants (80.7% vs. 72.8%). There was no statistically significant difference between the proportion of cisgender and non-binary participants born in Canada. Approximately 1 in 5 trans and non-binary participants were born in a country other than Canada.

Participants were able to select more than one sexual identity (i.e., selections were not mutually exclusive). There was a larger variation in sexual identities among trans and non-binary participants compared with cisgender participants (Table 1). Most cisgender participants identified as gay (85.2%) and/or bisexual (10.4%). More than one-third of trans respondents identified as gay (36.5%) and as queer (33.8%), and more than one-quarter as pansexual (27.4%) and as bisexual (27.0%). Compared with cisgender participants, trans participants were significantly less likely to identify as gay (36.5% vs. 85.2%), but significantly more likely to identify as queer (33.8% vs. 6.6%), pansexual (27.4% vs. 2.8%), bisexual (27.0% vs. 10.4%), asexual (3.7% vs. 0.6%), heteroflexible (2.4% vs. 0.6%), and other (2.4% vs. 0.4%). Half of

**Table 1. City of recruitment and demographic factors among cisgender, trans, and non-binary participants.**

| | Overall | | Cisgender | | Trans | | | Non-Binary | | |
|---|---|---|---|---|---|---|---|---|---|---|
| **City of Recruitment** | n | % | n | % | n | % | OR | n | % | OR |
| Calgary | 265 | 7.5% | 236 | 7.7% | 26 | 8.8% | Ref | 10 | 6.7% | Ref |
| Edmonton | 297 | 8.4% | 263 | 8.5% | 19 | 6.4% | 0.66 (0.35–1.21) | 15 | 10.0% | 1.35 (0.60–3.15) |
| Halifax | 199 | 5.6% | 164 | 5.3% | 27 | 9.1% | 1.45 (0.84–2.66) | 8 | 5.3% | 1.15 (0.43–2.98) |
| Kamloops and Kelowna | 121 | 3.4% | 101 | 3.3% | 18 | 6.1% | 1.62 (0.84–3.07) | 10 | 6.7% | 2.34 (0.93–5.86) |
| London | 154 | 4.4% | 115 | 3.7% | 30 | 10.1% | **2.37 (1.34–4.21)** | 11 | 7.3% | 2.26 (0.93–5.57) |
| Montreal | 418 | 11.9% | 351 | 11.4% | 26 | 8.8% | 0.67 (0.38–1.19) | 21 | 14.0% | 1.41 (0.67–3.18) |
| Ottawa | 411 | 11.7% | 371 | 12.0% | 24 | 8.1% | 0.59 (0.33–1.05) | 11 | 7.3% | 0.70 (0.29–1.70) |
| Toronto | 806 | 22.9% | 716 | 23.2% | 58 | 19.6% | 0.74 (0.46–1.21) | 33 | 22.0% | 1.09 (0.55–2.36) |
| Vancouver | 672 | 19.1% | 606 | 19.7% | 52 | 17.6% | 0.78 (0.48–1.29) | 23 | 15.3% | 0.90 (0.43–2.00) |
| Winnipeg | 181 | 5.1% | 160 | 5.2% | 16 | 5.4% | 0.91 (0.46–1.73) | 8 | 5.3% | 1.18 (0.44–3.06) |
| **Age Group** | | | | | | | | | | |
| <25 | 706 | 20.0% | 555 | 18.0% | 125 | 42.2% | Ref | 61 | 40.7% | Ref |
| 25–29 | 720 | 20.4% | 644 | 20.9% | 53 | 17.9% | **0.37 (0.26–0.51)** | 38 | 25.3% | **0.54 (0.35–0.81)** |
| 30–39 | 973 | 27.6% | 883 | 28.6% | 59 | 19.9% | **0.30 (0.21–0.41)** | 28 | 18.7% | **0.29 (0.18–0.45)** |
| 40–49 | 421 | 11.9% | 380 | 12.3% | 26 | 8.8% | **0.30 (0.19–0.47)** | 11 | 7.3% | **0.26 (0.13–0.49)** |
| 50–59 | 437 | 12.4% | 399 | 12.9% | 21 | 7.1% | **0.23 (0.14–0.67)** | 5 | 3.3% | **0.11 (0.04–0.26)** |
| 60+ | 230 | 6.5% | 206 | 6.7% | 9 | 3.0% | **0.19 (0.09–0.37)** | 5 | 3.3% | **0.22 (0.08–0.51)** |
| **Ethnoracial Identity (ref: no)** | | | | | | | | | | |
| African, Caribbean, Black | 145 | 4.1% | 124 | 4.0% | 14 | 4.7% | 1.18 (0.64–2.02) | 11 | 7.3% | 1.89 (0.94–3.43) |
| Arab, West Asian (e.g. Iranian, Afghan) | 116 | 3.3% | 101 | 3.3% | 8 | 2.7% | 0.82 (0.36–1.60) | 8 | 5.3% | 1.66 (0.73–3.28) |
| East or Southeast Asian (e.g. Chinese, Japanese, Korean) | 298 | 8.5% | 274 | 8.9% | 15 | 5.1% | **0.55 (0.31–0.90)** | 10 | 6.7% | 0.73 (0.36–1.34) |
| Indigenous | 314 | 8.9% | 250 | 8.1% | 49 | 16.6% | **2.25 (1.60–3.11)** | 24 | 16.0% | 0.73 (0.36–1.34) |
| Latin American, Hispanic | 180 | 5.1% | 159 | 5.2% | 13 | 4.4% | 0.84 (0.45–1.45) | 7 | 4.7% | **2.16 (1.34–3.34)** |
| South Asian (e.g. East Indian, Pakistani, Sri Lankan) | 113 | 3.2% | 89 | 2.9% | 14 | 4.7% | 1.67 (0.90–2.88) | 7 | 4.7% | 0.90 (0.38–1.82) |
| White | 2594 | 73.6% | 2271 | 73.7% | 227 | 76.7% | 1.18 (0.89–1.57) | 105 | 70.0% | 0.83 (0.59–1.20) |
| Other | 33 | 0.9% | 27 | 0.9% | 5 | 1.7% | 1.94 (0.66–4.68) | 4 | 2.7% | 3.10 (0.91–8.06) |
| **Born in Canada** | | | | | | | | | | |
| No | 883 | 25.1% | 799 | 25.9% | 52 | 17.6% | Ref | 30 | 20.0% | Ref |
| Yes | 2583 | 73.3% | 2245 | 72.8% | 239 | 80.7% | **1.64 (1.21–2.25)** | 115 | 76.7% | 1.36 (0.92–2.09) |
| **Sexual Identity (ref: no)** | | | | | | | | | | |
| Gay | 2816 | 79.9% | 2627 | 85.2% | 108 | 36.5% | **0.10 (0.07–0.12)** | 49 | 32.7% | **0.08 (0.06–0.12)** |
| Asexual | 33 | 0.9% | 19 | 0.6% | 11 | 3.7% | **6.22 (2.84–13.00)** | 7 | 4.7% | **8.00 (3.08–18.55)** |
| Straight | 27 | 0.8% | 20 | 0.6% | 5 | 1.7% | 2.63 (0.87–6.55) | 1 | 0.7% | 1.04 (0.06–5.05) |
| Bisexual | 426 | 12.1% | 320 | 10.4% | 80 | 27.0% | **3.20 (2.40–4.23)** | 26 | 17.3% | **1.84 (1.16–2.81)** |
| Pansexual | 178 | 5.1% | 87 | 2.8% | 81 | 27.4% | **13.00 (9.31–18.14)** | 48 | 32.0% | **16.57 (11.01–24.79)** |
| Queer | 330 | 9.4% | 204 | 6.6% | 100 | 33.8% | **7.22 (5.44–9.54)** | 74 | 49.3% | **14.19 (9.97–20.23)** |
| Heteroflexible | 27 | 0.8% | 17 | 0.6% | 7 | 2.4% | **4.37 (1.68–10.21)** | 2 | 1.3% | 2.47 (0.39–8.72) |
| Other | 21 | 0.6% | 13 | 0.4% | 7 | 2.4% | **5.72 (2.13–14.08)** | 4 | 2.7% | **6.47 (1.81–18.54)** |
| **Financial Strain** | | | | | | | | | | |
| Comfortable, with extra | 1364 | 38.7% | 1264 | 41.0% | 57 | 19.3% | **0.14 (0.09–0.21)** | 31 | 20.7% | **0.13 (0.07–0.21)** |
| Enough, but no extra | 1411 | 40.0% | 1244 | 40.4% | 118 | 39.9% | **0.29 (0.20–0.42)** | 53 | 35.3% | **0.22 (0.14–0.36)** |
| Have to cut back | 472 | 13.4% | 390 | 12.7% | 68 | 23.0% | **0.54 (0.36–0.81)** | 32 | 21.3% | **0.42 (0.25–0.72)** |
| Cannot make ends meet | 235 | 6.7% | 160 | 5.2% | 52 | 17.6% | Ref | 31 | 20.7% | Ref |

non-binary participants identified as queer (49.3%), and one-third as gay (32.7%) and as pansexual (32.0%). Compared with cisgender participants, non-binary participants were significantly less likely to identify as gay (32.7% vs. 85.2%), but significantly more likely to identify as queer (49.3% vs. 6.6%), pansexual (32.0% vs. 2.8%), bisexual (17.3% vs. 10.4%), asexual (4.7% vs. 0.6%), and other (2.7% vs. 0.4%). Non-binary participants were just as likely to identify as straight compared with cisgender participants (0.7% and 0.6% respectively).

Participants were asked to self-rate their money situation at one of four levels shown in the table below (Table 1). Overall, cisgender participants reported less financial strain than both trans and non-binary participants. More than 4 in 5 (81.4%) cisgender participants reported having enough money at the time of survey. Compared with cisgender participants, trans participants were approximately half as likely to report their money situation as "comfortable, with extra" (19.3% vs. 41.0%), almost twice as likely to report having to cut back (23.0% vs. 12.7%), and more than three times as likely to not be able to make ends meet (17.6% vs 5.2%). Compared with cisgender participants, non-binary participants were approximately half as likely to report their money situation as "comfortable, with extra" (20.7% vs. 41.0%), almost twice as likely to report having to cut back (21.3% vs. 12.7%), and four times as likely to report not being able to make ends meet (20.7% vs 5.2%).

Educational completion of participants was significantly different for both trans and non-binary participants when compared with cisgender participants (Table 2). Trans participants were more likely to report not completing high school (12.5% vs. 3.3%), only completing high school or equivalent (30.4% vs. 16.9%), and only completing post-secondary school (28.4% vs. 25.1%), and less likely to report completing a Bachelor's degree (19.3% vs. 32.2%) or above a bachelor's degree (9.5% vs. 22.1%) than cisgender participants. After adjusting for confounding variables, trans participants were still statistically significantly less likely to have completed a bachelor's degree or above a bachelor's degree. Non-binary participants were significantly less likely to report completing a Bachelor's degree (24.0% vs. 32.2%) or above a Bachelor's degree (8.0% vs. 22.1%), and significantly more likely to report completing post-secondary school (28.7% vs. 25.1%). However, none of these relationships proved statistically significant after adjusting for confounders. In the following tables, each variable was assessed in a separate model while adjusting for the same confounders.

Just less than half of participants in each group were single, with all three groups having approximately the same proportion of respondents who were single (45.6% of trans participants, 48.7% of non-binary participants, and 47.4% of cisgender participants) (Table 2). However, the gender of participants' partners varied significantly for trans and non-binary participants compared with cisgender participants. Approximately half as many trans and non-binary participants were in a relationship with a man (26.0% for trans and 22.0% for non-binary compared with 47.1% of cisgender participants). Trans respondents were significantly more likely than cisgender participants to be in a polyamorous relationship (11.5% vs. 1.8%), partnered with a woman (11.1% vs. 2.8%), or partnered with a non-binary person (4.1% vs 0.4%). After controlling for confounding variables, trans participants remained statistically more likely to be in polyamorous relationships compared with cisgender participants. Non-binary participants were also more likely than cisgender participants to be in a polyamorous relationship (16.7% vs. 1.8%), partnered with a woman (8.7% vs. 2.8%), or partnered with a non-binary person (4.0% vs 0.4%). Non-binary participants remained significantly less likely to be in a relationship with a man and significantly more likely to be in a polyamorous relationship after controlling for confounders when compared with cisgender participants. This question did not specify whether a man or woman partner was cisgender or transgender, which limits our understanding.

**Table 2. Education, relationship status, mental health, and substance use among cisgender, trans, and non-binary participants.**

| | Overall | | Cisgender | | Trans | | | | Non-binary | | | |
|---|---|---|---|---|---|---|---|---|---|---|---|---|
| **Level of education** | **n** | **%** | **n** | **%** | **n** | **%** | **OR** | **AOR** | **n** | **%** | **OR** | **AOR** |
| Did not finish high school | 155 | 4.4% | 103 | 3.3% | 37 | 12.5% | Ref | Ref | 12 | 8.0% | Ref | Ref |
| High school or equivalent | 646 | 18.3% | 521 | 16.9% | 90 | 30.4% | **0.48 (0.31–0.75)** | 0.63 (0.38–1.06) | 47 | 31.3% | 0.77 (0.41–1.57) | 1.53 (0.69–3.67) |
| Post-secondary school (e.g. certificate, diploma) | 897 | 25.5% | 775 | 25.1% | 84 | 28.4% | **0.30 (0.20–0.47)** | 0.75 (0.45–1.27) | 43 | 28.7% | **0.48 (0.25–0.97)** | 1.64 (0.73–4.03) |
| Bachelor's degree | 1079 | 30.6% | 994 | 32.2% | 57 | 19.3% | **0.16 (0.10–0.25)** | **0.50 (0.29–0.87)** | 36 | 24.0% | **0.31 (0.16–0.64)** | 1.53 (0.66–3.83) |
| Above a bachelor's degree (e.g., masters, doctorate) | 724 | 20.5% | 680 | 22.1% | 28 | 9.5% | **0.11 (0.07–0.19)** | **0.46 (0.24–0.87)** | 12 | 8.0% | **0.15 (0.07–0.35)** | 0.79 (0.28–2.30) |
| **Relationship Status** | | | | | | | | | | | | |
| No | 1661 | 47.1% | 1460 | 47.4% | 135 | 45.6% | Ref | Ref | 73 | 48.7% | Ref | Ref |
| Yes, with a man | 1580 | 44.8% | 1452 | 47.1% | 77 | 26.0% | **0.57 (0.43–0.76)** | 0.86 (0.62–1.18) | 33 | 22.0% | **0.45 (0.30–0.68)** | **0.62 (0.38–0.98)** |
| Yes, with a woman | 126 | 3.6% | 87 | 2.8% | 33 | 11.1% | **4.10 (2.62–6.30)** | 1.07 (0.65–1.73) | 13 | 8.7% | **2.99 (1.53–5.43)** | 0.75 (0.36–1.43) |
| Yes, with a non-binary person | 26 | 0.7% | 12 | 0.4% | 12 | 4.1% | **10.81 (4.72–24.79)** | 2.70 (1.00–7.32) | 6 | 4.0% | **10.00 (3.40–26.52)** | 1.79 (0.44–6.21) |
| Yes, with more than 1 person (polyamorous) | 96 | 2.7% | 55 | 1.8% | 34 | 11.5% | **6.69 (4.18–10.57)** | **3.98 (2.30–6.85)** | 25 | 16.7% | **9.09 (5.30–15.30)** | **5.14 (2.69–9.70)** |
| **Depression Scores** | | | | | | | | | | | | |
| < Score of 3 | 2774 | 78.7% | 2500 | 81.1% | 182 | 61.5% | Ref | Ref | 88 | 58.7% | Ref | Ref |
| ≥ Score of 3 | 502 | 14.2% | 386 | 12.5% | 89 | 30.1% | **3.17 (2.40–4.16)** | **1.77 (1.27–2.43)** | 50 | 33.3% | **3.68 (2.54–5.27)** | **2.00 (1.30–3.04)** |
| **Anxiety Scores** | | | | | | | | | | | | |
| < Score of 3 | 2643 | 75.0% | 2404 | 78.0% | 155 | 52.4% | Ref | Ref | 74 | 49.3% | Ref | Ref |
| ≥ Score of 3 | 628 | 17.8% | 480 | 15.6% | 115 | 38.9% | **3.72 (2.86–4.82)** | **2.11 (1.56–2.86)** | 63 | 42.0% | **4.26 (3.00–6.05)** | **2.37 (1.57–3.55)** |
| **Help Wanted With Mental Health Issues (ref: no)** | | | | | | | | | | | | |
| Depression | 820 | 23.3% | 651 | 21.1% | 140 | 47.3% | **3.60 (2.79–4.65)** | **2.26 (1.69–3.01)** | 61 | 40.7% | **2.54 (1.79–3.58)** | **1.55 (1.04–2.29)** |
| Anxiety | 984 | 27.9% | 801 | 26.0% | 146 | 49.3% | **2.98 (2.31–3.84)** | **2.15 (1.61–2.88)** | 69 | 46.0% | **2.42 (1.72–3.41)** | **1.87 (1.27–2.76)** |
| Gender dysphoria/transition | 123 | 3.5% | 17 | 0.6% | 100 | 33.8% | **96.76 (58.04–171.01)** | **44.32 (25.19–82.07)** | 48 | 32.0% | **85.32 (48.17–158.05)** | **45.46 (22.87–94.27)** |
| Eating disorders | 195 | 5.5% | 132 | 4.3% | 54 | 18.2% | **5.07 (3.57–7.13)** | **2.80 (1.84–4.22)** | 30 | 20.0% | **5.51 (3.50–8.46)** | **2.85 (1.66–4.80)** |
| Body image | 621 | 17.6% | 497 | 16.1% | 108 | 36.5% | **3.10 (2.38–4.03)** | **2.30 (1.69–3.12)** | 56 | 37.3% | **3.09 (2.16–4.37)** | **2.19 (1.46–3.27)** |
| Suicidal thoughts | 240 | 6.8% | 171 | 5.5% | 62 | 20.9% | **4.60 (3.31–6.32)** | **2.30 (1.56–3.35)** | 24 | 16.0% | **3.18 (1.95–4.98)** | 1.32 (0.75–2.25) |
| **Mental Health resources used, past year (ref: no)** | | | | | | | | | | | | |
| Elder (Indigenous) | 83 | 2.4% | 60 | 1.9% | 16 | 5.4% | **2.90 (1.59–4.98)** | 1.95 (0.95–3.84) | 11 | 7.3% | **3.86 (1.88–7.23)** | 2.00 (0.82–4.51) |
| Knowledge Keeper (Indigenous) | 51 | 1.4% | 36 | 1.2% | 13 | 4.4% | **3.91 (1.98–7.29)** | **3.64 (1.61–7.85)** | 9 | 6.0% | **5.23 (2.32–10.63)** | **3.46 (1.24–8.79)** |
| Psychiatrist | 414 | 11.7% | 320 | 10.4% | 78 | 26.4% | **3.19 (2.38–4.24)** | **1.80 (1.29–2.50)** | 38 | 25.3% | **2.86 (1.91–4.19)** | 1.59 (1.00–2.46) |
| Clinical psychologist | 325 | 9.2% | 269 | 8.7% | 47 | 15.9% | **2.01 (1.42–2.80)** | **1.67 (1.12–2.44)** | 25 | 16.7% | **2.04 (1.28–3.16)** | 1.71 (1.00–2.81) |

(*Continued*)

**Table 2.** (Continued)

| Level of education | Overall | | Cisgender | | Trans | | | | Non-binary | | | |
|---|---|---|---|---|---|---|---|---|---|---|---|---|
| | n | % | n | % | n | % | OR | AOR | n | % | OR | AOR |
| Registered counsellor | 486 | 13.8% | 378 | 12.3% | 91 | 30.7% | **3.30 (2.50–4.35)** | **2.28 (1.66–3.12)** | 49 | 32.7% | **3.42 (2.36–4.90)** | **2.46 (1.61–3.71)** |
| Peer counsellor/Navigator | 179 | 5.1% | 124 | 4.0% | 47 | 15.9% | **4.60 (3.17–6.57)** | **3.71 (2.41–5.67)** | 23 | 15.3% | **4.21 (2.54–6.70)** | **2.94 (1.63–5.12)** |
| Social worker | 303 | 8.6% | 209 | 6.8% | 75 | 25.3% | **4.83 (3.56–6.52)** | **2.50 (1.74–3.57)** | 45 | 30.0% | **5.83 (3.95–8.50)** | **3.05 (1.91–4.80)** |
| None of the above | 1984 | 56.3% | 1826 | 59.2% | 92 | 31.1% | **0.27 (0.21–0.35)** | **0.44 (0.33–0.60)** | 48 | 32.0% | **0.27 (0.19–0.38)** | **0.46 (0.31–0.70)** |
| **Any Substance Use, Past 6 Months** | | | | | | | | | | | | |
| Yes | 2580 | 73.2% | 2272 | 73.7% | 219 | 74.0% | Ref | Ref | 113 | 75.3% | Ref | Ref |
| No | 658 | 18.7% | 582 | 18.9% | 48 | 16.2% | 1.17 (0.85–1.63) | 0.97 (0.67–1.42) | 27 | 18.0% | 1.07 (0.71–1.68) | 0.90 (0.56–1.52) |

## Mental health and substance use

When trans and non-binary participants were compared to cisgender participants, statistically significant differences in depression scores emerged (Table 2). Nearly a third of trans (30.1%) and non-binary participants (33.3%) scored greater than or equal to 3, compared with only 1 in 8 (12.5%) cisgender participants. These results remained statistically significant after adjusting for confounding variables. Similarly, statistically significant differences were found when anxiety scores where compared across groups (Table 2). Approximately 2 in 5 trans (38.9%) and non-binary participants (42.0%) scored a 3 or higher on the GAD-2, compared to 15.6% of cis participants. After adjusting for confounding variables, differences in anxiety scores remained statistically significant for both trans and non-binary participants.

When participants were asked if they wanted help with a variety of mental health issues, there were statistically significant differences between both trans and non-binary participants and cisgender participants on almost every issue (Table 2). Trans and non-binary participants were both significantly more likely than cisgender participants to indicate they wanted help with depression and anxiety. Nearly half of trans respondents wanted help for depression (47.3%) and anxiety (49.3%). About 2 in 5 non-binary participants wanted help for depression (40.7%) and nearly half for anxiety (46.0%). In contrast, roughly one-fifth (21.1%) of cisgender participants indicated the same for depression and 1 in 4 (26.0%) indicated the same for anxiety. Differences in depression and anxiety scores for trans and non-binary participants in comparison to cisgender participants remained statistically significant after adjusting for confounders.

Besides depression and anxiety, the most common issue with which participants wanted help was body image, regardless of gender identity, with 36.5% of trans participants and 37.3% of non-binary participants reporting this—more than double that of cisgender participants (16.1%). Highly statistically significant results were observed for help wanted with gender dysphoria/transition, with approximately a third of trans (33.8%) and non-binary (32.0%) participants indicating wanting help with this compared to 0.6% of cisgender participants. Statistically significant differences also resulted when comparing help wanted with eating disorders (18.2% of trans and 20.0% of non-binary participants vs. 4.3% of cisgender participants) and suicidal thoughts (20.9% of trans and 16.0% of non-binary participants vs. 5.5% of cisgender participants). After adjusting for confounding variables, all these results remained

statistically significant with the exception of findings about suicidal thoughts when comparing non-binary and cisgender participants.

In general, significantly more trans participants and non-binary participants used resources for health in the past year than cisgender participants (Table 2). Less than a third of trans participants (31.1%) and non-binary participants (32.0%) reported not having used any of the health resource in the past year compared with 3 in 5 (59.2%) cisgender respondents. Trans participants and non-binary participants were approximately three times more likely than Indigenous cisgender participants to access support from an Indigenous Elder, and approximately four times more likely to visit an Indigenous Knowledge Keeper (five times more likely for non-binary participants). Recall that compared with cisgender participants, twice as many Indigenous participants were trans and non-binary. With the exception of trans participants' access to Elders, these results remained statistically significant after adjusting for confounding variables.

Compared with cisgender participants, trans participants were more likely to have gone to a registered counsellor (30.7% vs. 12.3%), a psychiatrist (26.4% vs. 10.4%), a social worker (25.3% vs. 6.8%), a clinical psychologist (15.9% vs. 8.7%) and a peer counsellor/navigator (15.9% vs. 4.0%). All these results were statistically significant after adjusting for confounding variables. Compared with cisgender participants, non-binary participants were more likely to have gone to a registered counsellor (32.7% vs. 12.3%), a social worker (30.0% vs. 6.8%), a psychiatrist (25.3% vs. 10.4%), a clinical psychologist (16.7% vs. 8.7%) and a peer counsellor/navigator (15.3% vs. 4.0%). However, the results for psychiatrist and clinical psychologist were no longer statistically significant after adjusting for confounding variables.

Overall, there were no statistically significant differences across gender identity groups when asked if they had used any substances (i.e., alcohol or drugs) in the past 6 months (Table 2). This was reported by approximately three quarters of trans participants (74.0%), non-binary participants (75.3%), and cisgender participants (73.7%).

## Sexual health

Participants were asked a series of questions about their sexual health, including whether they had had chlamydia, gonorrhea, or syphilis in the past year (Table 3). Trans participants were significantly less likely to report having had one of these STIs in the past year when compared with cisgender participants (9.8% vs. 16.4%), however this was no longer statistically significant after adjusting for confounding variables. There was no significant difference between non-binary and cisgender participants in their reporting of these STIs in the past year (12.0% vs. 16.4%).

When compared with cisgender participants, trans and non-binary participants were less likely to have been tested for STIs in the past year (Table 3). About 3 in 5 trans (59.8%) and non-binary participants (62.0%) had been tested in past year, compared with 68.5% of cisgender participants. Trans participants (15.5%) and non-binary participants (13.3%) were approximately twice as likely to report never having been tested for STIs compared to cisgender participants (7.0%). These findings were no longer statistically significant after adjusting for confounders. There were no significant differences when comparing non-binary and cisgender participants. Participants were asked to report whether various issues had caused them to delay or skip STI testing over the past year. Across all participants, the most commonly reported issues encountered were being too busy (25.1%), hours being inconvenient (15.4%), and feeling "stressed out, anxious, or depressed" (10.4%).

Generally, trans and non-binary participants were less likely to have had various samples collected during their most recent STI test (Table 3). When compared with cisgender

**Table 3. Sexual health, Hepatitis C, and HIV among cisgender, trans, and non-binary participants.**

| | Overall | | Cisgender | | Trans | | | | Non-Binary | | | |
|---|---|---|---|---|---|---|---|---|---|---|---|---|
| **Chlamydia, gonorrhea, or syphilis, past year** | **n** | **%** | **n** | **%** | **n** | **%** | **OR** | **AOR** | **n** | **%** | **OR** | **AOR** |
| Yes | 559 | 15.9% | 507 | 16.4% | 29 | 9.8% | **0.62 (0.41–0.91)** | 0.89 (0.57–1.36) | 18 | 12.0% | 0.74 (0.43–1.21) | 1.04 (0.58–1.80) |
| No | 2540 | 72.1% | 2244 | 72.8% | 208 | 70.3% | Ref | Ref | 107 | 71.3% | Ref | Ref |
| **Timing of Last STI Test** | | | | | | | | | | | | |
| In the past year | 2373 | 67.3% | 2113 | 68.5% | 177 | 59.8% | Ref | Ref | 93 | 62.0% | Ref | Ref |
| Longer than a year ago | 691 | 19.6% | 619 | 20.1% | 48 | 16.2% | 0.93 (0.66–1.28) | 0.92 (0.63–1.33) | 25 | 16.7% | 0.92 (0.57–1.42) | 0.86 (0.50–1.42) |
| Never | 278 | 7.9% | 216 | 7.0% | 46 | 15.5% | **2.54 (1.77–3.59)** | 1.05 (0.68–1.58) | 20 | 13.3% | **2.10 (1.24–3.41)** | 0.76 (0.41–1.34) |
| **Samples collected during last STI test (ref: no)** | | | | | | | | | | | | |
| Urine test | 2438 | 69.2% | 2196 | 71.2% | 172 | 58.1% | **0.73 (0.54–0.99)** | **0.69 (0.49–0.99)** | 91 | 60.7% | 0.77 (0.51–1.82) | 0.79 (0.49–1.31) |
| Blood sample | 2872 | 81.5% | 2569 | 83.3% | 205 | 69.3% | **0.56 (0.37–0.86)** | 0.95 (0.59–1.56) | 109 | 72.7% | 0.66 (0.38–1.25) | 1.33 (0.69–2.80) |
| Throat swab | 1521 | 43.2% | 1391 | 45.1% | 87 | 29.4% | **0.59 (0.44–0.77)** | 0.77 (0.56–1.04) | 46 | 30.7% | **0.60 (0.41–0.87)** | 0.82 (0.54–1.24) |
| Rectal swab (in your bum) | 1151 | 32.7% | 1055 | 34.2% | 63 | 21.3% | **0.60 (0.44–0.80)** | 0.87 (0.62–1.21) | 36 | 24.0% | 0.68 (0.45–1.00) | 1.09 (0.69–1.69) |
| None of the above | 85 | 2.4% | 67 | 2.2% | 8 | 2.7% | 1.43 (0.63–2.84) | 1.02 (0.42–2.24) | 5 | 3.3% | 1.73 (0.60–3.96) | 1.00 (0.28–2.81) |
| **Ever Tested for HCV** | | | | | | | | | | | | |
| Yes | 2507 | 71.1% | 2240 | 72.7% | 179 | 60.5% | Ref | Ref | 94 | 62.7% | Ref | Ref |
| No | 327 | 9.3% | 263 | 8.5% | 53 | 17.9% | **2.52 (1.80–3.50)** | 1.20 (0.79–1.77) | 18 | 12.0% | 1.63 (0.94–2.68) | 0.75 (0.39–1.36) |
| **Ever diagnosed with HCV** | | | | | | | | | | | | |
| No | 2431 | 69.0% | 2211 | 71.7% | 165 | 55.7% | Ref | Ref | 87 | 58.0% | Ref | Ref |
| Yes | 33 | 0.9% | 26 | 0.8% | 4 | 1.4% | 2.14 (0.63–5.60) | 1.08 (0.28–3.41) | 3 | 2.0% | 3.05 (0.72–8.91) | 2.43 (0.46–9.85) |
| **Ever Tested for HIV** | | | | | | | | | | | | |
| Yes | 3004 | 85.2% | 2691 | 87.3% | 210 | 70.9% | Ref | Ref | 109 | 72.7% | Ref | Ref |
| No | 296 | 8.4% | 230 | 7.5% | 48 | 16.2% | **2.67 (1.88–3.73)** | 0.88 (0.58–1.31) | 23 | 15.3% | **2.47 (1.51–3.88)** | 0.80 (0.45–1.36) |
| **Ever Diagnosed with HIV** | | | | | | | | | | | | |
| No | 2745 | 77.9% | 2459 | 79.8% | 193 | 65.2% | Ref | Ref | 103 | 68.7% | Ref | Ref |
| Yes | 225 | 6.4% | 200 | 6.5% | 15 | 5.1% | 0.96 (0.53–1.59) | 1.43 (0.75–2.56) | 8 | 5.3% | 3.05 (0.72–8.91) | 0.68 (0.65–3.81) |
| **Ever used HIV PrEP** | | | | | | | | | | | | |
| No | 2649 | 75.2% | 2300 | 74.6% | 246 | 83.1% | Ref | Ref | 123 | 82.0% | Ref | Ref |
| Yes, but I stopped | 96 | 2.7% | 80 | 2.6% | 13 | 4.4% | 1.52 (0.80–2.68) | **2.96 (1.45–5.65)** | 5 | 3.3% | 1.17 (0.41–2.66) | 2.34 (0.73–6.06) |
| Yes, I'm taking PrEP now! | 400 | 11.4% | 379 | 12.3% | 12 | 4.1% | **0.30 (0.16–0.51)** | 0.56 (0.28–1.00) | 7 | 4.7% | **0.35 (0.15–0.69)** | 0.73 (0.29–1.53) |

participants, trans participants were significantly less likely to have completed a urine test (58.1% vs. 71.2%), blood sample (69.3% vs. 83.3%), throat swab (29.4% vs. 45.1%), or rectal swab (21.3% vs. 34.2%) during their most recent STI test. Most of these relationships were no longer statistically significant after adjusting for confounding variables, but trans participants remained significantly less likely to have completed a urine test after these adjustments were

made. When compared with cisgender participants, the only statistically significant finding for non-binary participants was that they were less likely to have completed a throat swab (30.7% vs. 45.1%), although they were also less likely to have completed a urine test (60.7% vs. 71.2%), blood sample (72.7% vs. 83.3%), or rectal swab (24.0% vs. 34.2%).

When asked if they had ever tested for hepatitis C, trans participants were significantly less likely to have ever tested compared to cisgender participants (Table 3). Twice as many trans participants (17.9%) reported never having tested when compared to cisgender participants (8.5%). This relationship was no longer statistically significant after adjusting for confounders. Non-binary participants were also more likely to have never tested for hepatitis C (12.0%) when compared with cisgender participants (8.5%), but this was not a statistically significant relationship. Participants were asked if they had ever been diagnosed with hepatitis C (Table 3). Generally, rates of hepatitis C diagnosis were low across the sample with no statistically significant differences across gender groups. However, trans participants were nearly twice as likely as cisgender participants to report a diagnosis (1.4% vs. 0.8%), while non-binary participants were 2.5 times more likely to report a diagnosis (2.0%).

When asked if they had ever been tested for HIV, trans and non-binary participants were significantly more likely to report never having been tested compared to cisgender participants (Table 3). Over twice as many trans (16.2%) and non-binary (15.3%) participants reported never having been tested compared to cisgender participants (7.5%). Neither relationship was statistically significant after adjusting for confounding variables. Prevalence of HIV diagnosis across gender groups were similar, with no statistically significant differences (Table 3).

Participants who were not living with HIV were asked if they had ever used PrEP and whether they were currently using it (Table 3). Trans and non-binary participants were significantly less likely (2 to 3 times) to report currently using PrEP when compared with cisgender participants. Approximately 1 in 25 trans participants (4.1%) and 1 in 20 non-binary participants (4.7%) were currently using PrEP, compared to 1 in 8 cisgender participants. Adjusting for confounders rendered these findings no longer statistically significant. Trans participants were also more likely to report having started using PrEP and stopped when compared with cisgender participants (4.4% vs. 2.6%). This became a statistically significant finding after adjusting for confounding variables. Participants who were not using PrEP were asked to report what stops them from taking PrEP. Across all groups, the most common reasons reported were "I don't think I'll get HIV" (25.8%), cost (19.8%), side effects (14.7%), and none of the above (15.9%).

## Healthcare access

Participants were asked a series of questions about healthcare access, including if they had a regular doctor or nurse practitioner (Table 4). There were no statistically significant differences in access to a regular healthcare provider when non-binary participants were compared with cisgender participants (71.3% vs. 71.7%). However, trans participants were more likely to have a regular doctor or nurse practitioner when compared with cisgender participants (76.4% vs. 71.7%). This became a statistically significant finding after adjusting for confounding variables.

When asked if they had ever been vaccinated against hepatitis B, no statistically significant findings emerged when comparing trans and non-binary participants to cisgender participants (Table 4). Trans participants were more likely to indicate never having been vaccinated (13.5% vs. 11.9%) or being unsure of their vaccination status (22.3% vs. 14.9%) in comparison to cisgender participants. Similarly, non-binary participants were more likely to indicate never

**Table 4. Healthcare access, vaccinations, among cisgender, trans, and non-binary participants.**

| | Overall | | Cisgender | | Trans | | | | Non-Binary | | | |
|---|---|---|---|---|---|---|---|---|---|---|---|---|
| **Has regular HCP** | n | % | n | % | n | % | OR | AOR | n | % | OR | AOR |
| No | 851 | 24.1% | 752 | 24.4% | 59 | 19.9% | Ref | Ref | 40 | 26.7% | Ref | Ref |
| Yes | 2524 | 71.6% | 2212 | 71.7% | 226 | 76.4% | 1.30 (0.97–1.77) | **1.82 (1.30–2.57)** | 107 | 71.3% | 0.91 (0.63–1.33) | 1.53 (1.00–2.38) |
| **Ever vaccinated for HBV** | | | | | | | | | | | | |
| No | 435 | 12.3% | 367 | 11.9% | 40 | 13.5% | Ref | Ref | 21 | 14.0% | Ref | Ref |
| Unsure | 546 | 15.5% | 459 | 14.9% | 66 | 22.3% | 1.32 (0.87–2.01) | 1.23 (0.77–1.99) | 34 | 22.7% | 1.29 (0.74–2.30) | 1.18 (0.63–2.24) |
| Yes | 2336 | 66.3% | 2092 | 67.9% | 173 | 58.4% | 0.76 (0.53–1.10) | 1.15 (0.77–1.76) | 88 | 58.7% | 0.74 (0.46–1.23) | 1.09 (0.64–1.94) |
| **Ever vaccinated for HPV** | | | | | | | | | | | | |
| No | 1361 | 38.6% | 1221 | 39.6% | 86 | 29.1% | Ref | Ref | 41 | 27.3% | Ref | Ref |
| Unsure | 709 | 20.1% | 605 | 19.6% | 75 | 25.3% | **1.76 (1.27–2.43)** | 1.21 (0.83–1.74) | 38 | 25.3% | **1.87 (1.19–2.94)** | 1.29 (0.77–2.14) |
| Yes | 1083 | 30.7% | 942 | 30.6% | 113 | 38.2% | **1.70 (1.27–2.29)** | **1.51 (1.08–2.12)** | 54 | 36.0% | **1.71 (1.13–2.60)** | 1.47 (0.92–2.36) |
| **Asked and Denied (ref: no)** | | | | | | | | | | | | |
| An HIV test | 124 | 3.5% | 103 | 3.3% | 17 | 5.7% | 1.75 (1.00–2.89) | 1.29 (0.68–2.30) | 9 | 6.0% | 1.85 (0.85–3.55) | 1.40 (0.60–2.94) |
| PEP | 42 | 1.2% | 32 | 1.0% | 8 | 2.7% | **2.61 (1.11–5.46)** | 2.29 (0.87–5.42) | 3 | 2.0% | 1.95 (0.46–5.52) | 1.69 (0.35–5.77) |
| PrEP | 80 | 2.3% | 72 | 2.3% | 7 | 2.4% | 1.00 (0.41–2.04) | 0.99 (0.37–2.25) | 3 | 2.0% | 0.85 (0.21–2.32) | 0.87 (0.19–2.67) |
| HPV vaccination | 62 | 1.8% | 49 | 1.6% | 12 | 4.1% | **2.58 (1.30–4.76)** | **2.78 (1.25–5.79)** | 5 | 3.3% | 2.14 (0.73–4.97) | 1.87 (0.56–5.19) |
| Hormone therapy | 57 | 1.6% | 9 | 0.3% | 46 | 15.5% | **63.13 (32.00–139.36)** | **31.60 (14.43–75.96)** | 20 | 13.3% | **53.26 (24.41–125.38)** | **24.82 (9.19–72.61)** |
| Gender affirming surgery | 42 | 1.2% | 5 | 0.2% | 36 | 12.2% | **84.95 (36.15–249.04)** | **50.65 (19.10–161.85)** | 14 | 9.3% | **63.92 (24.04–200.38)** | **36.71 (10.69–144.38)** |
| None of the above | 2891 | 82.0% | 2595 | 84.2% | 192 | 64.9% | **0.20 (0.15–0.26)** | **0.24 (0.17–0.34)** | 102 | 68.0% | **0.24 (0.16–0.37)** | **0.28 (0.17–0.46)** |

having been vaccinated (14.0% vs. 11.9%) or being unsure of their vaccination status (22.7% vs. 14.9%).

However, statistically significant differences emerged in terms of HPV vaccination upon comparison across gender groups (Table 4). Trans and non-binary participants were less likely to report never having been vaccinated when compared with cisgender participants (29.1% of trans participants, 27.3% of non-binary participants, and 39.6% of cisgender participants). Compared with cisgender participants, trans participants were significantly more likely to indicate being vaccinated against HPV (38.2% vs. 30.6%) and being unsure of their vaccination status (25.3% vs. 19.6%). After adjusting for confounding variables, trans participants were still significantly more likely to report being vaccinated for HPV. Non-binary participants were also significantly more likely to indicate being vaccinated against HPV (36.0% vs. 30.6%) and being unsure of their vaccination status (25.3% vs. 19.6%) in comparison to cisgender participants. However, these findings were not statistically significant after adjusting for confounders.

Finally, participants were asked if they had ever asked for *and* been denied a variety of health services (Table 4). Trans and non-binary participants were both significantly less likely to report having never been denied one of the listed services in comparison to cisgender participants (64.9% of trans participants, 68.0% of non-binary participants, and 84.2% of cisgender participants), and this remained statistically significant after adjusting for confounding variables. Compared to cisgender participants, trans participants were also significantly more likely to report being denied PEP [post-exposure prophylaxis] (2.7% vs 1.0%), HPV vaccination (4.1% vs. 1.6%), hormone therapy (15.5% vs. 0.3%), and gender affirming surgery (12.2%

vs. 0.2%). Results remained statistically significant for all these health services after adjusting for confounding variables, with the exception of being denied PEP. Non-binary participants were significantly more likely to report being denied hormone therapy (13.3% vs. 0.3%) and gender affirming surgery (9.3% vs. 0.2%) when compared with cisgender participants, and both relationships remained highly statistically significant after adjusting for confounders.

## Discussion

This study aimed to illuminate the health and socio-demographic characteristics of transgender and non-binary populations in Canada. Using data from Sex Now 2018, we compared cisgender participants with trans participants and with non-binary participants across a number of characteristics. We begin with a discussion of healthcare related differences followed by mental health and sociodemographic drivers. Where possible, throughout this discussion, literature specific to Canada is used to give context to and support our results.

Our current study highlights a number of important differences in trans and non-binary people's experiences and needs with respect to general healthcare and more specifically sexual health. Trans people reported fewer STIs over the last year and less testing for STIs. Significantly, they were more likely to never have been tested for HIV, which reflects existing evidence that trans men test less frequently for HIV and STIs [6–8]. Although we did not report on participants' sexual behaviours, existing literature also suggests lower prevalence of high-risk sexual encounters among trans men [6, 7], which may contribute to lower levels of perceived risk and therefore reduced rates of HIV and STI testing. Indeed, when asked why they did not use PrEP, trans and non-binary participants were more likely to report perceived low risk for getting HIV. This low risk perception may be due to engaging in sexual behaviours that are considered low risk, utilizing other strategies to reduce risk, or engaging in sex with people they know to be HIV negative [15]. However, low rates of testing among trans and non-binary participants may also be indicative of participants' discomfort in accessing sexual health services in gendered spaces (e.g. women's clinic or gay men's health clinic). In addition, they could avoid testing or other sexual healthcare due to fears of mistreatment by healthcare providers, as other studies have found [16].

Trans participants reported higher rates of access to a regular healthcare provider when compared to the cisgender group, while non-binary participants reported access at similar rates to cisgender participants. Despite this, differences likely exist in the quality of care received by trans and non-binary people. Several other studies have highlighted gaps in healthcare provider competence for trans people's health [17, 18]. Others found that trans respondents need to educate their healthcare providers in order to receive appropriate care [19, 20]. These experiences may lead trans and non-binary people to delay care or not access certain services. Indeed, the health of trans people is likely impacted by challenges with accessing gender-affirming care [21]. In the present study, we did not specifically ask trans respondents whether they felt their healthcare provider was trans competent or affirming of their gender and, therefore, cannot draw related conclusions. There is a need for more research specific to healthcare access and quality for trans and non-binary people who belong to sexual minority groups.

Regarding mental health, statistically significant differences existed in the depression and anxiety scores of trans and non-binary participants when compared with cis participants, with trans and non-binary people reporting higher rates of depression and anxiety. This adds to a body of literature that demonstrates that trans people face mental health challenges at much higher rates [2, 22] than cisgender people [23]. Further, sexual minorities are more likely than heterosexual people to experience mental health challenges that include depression, anxiety,

and suicidality [24]. For our trans and non-binary participants who were also sexual minorities, these multiple identities may increase the likelihood of experiencing depression or anxiety. Additional factors contribute to poor mental health within trans and non-binary communities, including transphobia, discrimination, and stigma, which reduce overall well-being and increase suicidality [25, 26]. Trans and non-binary people reported wanting help for depression and anxiety more than cisgender participants and used resources for mental health more often. Despite wanting and utilizing more mental health services, trans and non-binary people still experienced poorer mental health. Few trans-specific mental health services may be available to our sample [2], and those that do exist may be unsatisfactory and/or ineffective at meeting trans and non-binary people's needs. A comprehensive mental health research strategy and response plan is recommended to address health inequities experienced by these communities. Future research and evaluation efforts must carefully study the effectiveness of these resources and their ability to address the unique needs and desires of trans and non-binary people.

Our sample of trans and non-binary participants had several notable differences in sociodemographic drivers of healthcare access and outcomes, which both support and diverge from existing literature about the demographics of trans and non-binary populations. The proportion of trans and non-binary people in our sample was high compared with the general population [27], especially for those with non-binary or genderfluid identities, which may reflect the demographic composition of Pride festival attendees. Similarly, TransPulse Ontario found about 40% of their trans sample identified as 'fluid' or having a non-binary gender identity [28]. Surprisingly, we did not recruit a significantly larger number of trans and non-binary people in major cities, where more healthcare services targeted for trans people typically exist, and more liberal political landscapes may provide policies that support trans rights. Our sample of trans and non-binary people was significantly younger than our cisgender sample, which may be due to our recruitment occurring through Pride festivals, which younger people may be more likely to attend. Some literature points to a recent trend of more people coming out as trans or non-binary, and at younger ages; this may explain the skew of our sample toward younger participants [29]. Research and programs need better strategies to reach and engage with older trans and non-binary people, especially those who may not identify with "LGBTQ2S+" or sexual orientation minority groups, spaces or places, as older trans and non-binary people may have different experiences and needs.

Trans respondents were more likely to be born in Canada than cisgender participants; this may reflect challenges trans people face in immigrating to Canada. Future work should examine related policies and the immigration experiences of trans people in more detail since it is outside the scope of this analysis. The results of this study add to growing literature about sexual orientation among trans and non-binary people. Trans people in our sample were significantly more likely to identify as queer, and many used pansexual or bisexual to describe their sexual orientation. This corroborates findings from TransPulse Canada, whose participants mostly did not self-identify as heterosexual (8% reported being heterosexual) [5]. We found trans and non-binary people to be single at the same rates as cisgender participants; however, trans and non-binary people were significantly more likely to be in polyamorous relationships. These relationships may provide unique opportunities, challenges, and strengths for trans and non-binary people that should not be ignored. Future research, policy, and programs should not assume or require monogamy and should affirm diverse sexual partnership formations.

Our participants' reports of educational attainment corroborate some existing studies [30], as trans and non-binary participants generally received less post-secondary education than cisgender participants. Other studies show higher educational attainment among trans people, yet higher rates of unemployment and low income despite this [31]. This likely reflects the

younger age composition of our trans and non-binary participants. In terms of finances, trans and non-binary participants reported being less financially comfortable than their cisgender counterparts. In Canada in 2018, 12.5% of the general population met the Statistics Canada threshold for being low income [32]. Our survey did not ask whether participants met the criteria for being considered low income, however, 17% of trans and 21% of non-binary people reported not being able to make ends meet. Our findings echo the results of several other studies which demonstrate that trans people are more likely to be low income or unemployed. For example, among TransPulse Canada respondents, 26% had an income of less than $15,000 a year [5]. These economic barriers may impact the overall health and well-being of this population [17]. Greater supports throughout primary, secondary, and tertiary education could help reduce socioeconomic disparities experienced by trans and non-binary individuals. Further, stronger and more active employment protections for trans and non-binary people are needed to ensure equity in hiring processes, workplace environments, and compensation. Future research should examine educational goals, experiences, and retention in programs, and generally explore what kinds of efforts to increase formal education and training may be necessary and desired among trans and non-binary communities.

## Limitations and future research

Our findings show that differences in health outcomes exist for cisgender, transgender, and non-binary people in Canada, drawn from a survey of GBT2Q+ people. While our study adds to a growing body of evidence that trans people have unique health and healthcare concerns, it is not without limitations. The Sex Now survey is not a comprehensive study of all issues affecting trans and non-binary people in Canada and did not collect some information that may have enhanced the conclusions we can draw from this analysis. Furthermore, the Sex Now study is generally known and promoted as a survey of "sex between guys," although more detailed language and specific eligibility was available in the current survey cycle. This men-focussed approach erases non-binary people, and some potentially eligible trans and non-binary people may have not participated because the language was not inclusive. Future surveys of sexual and gender minority populations should be more intentional about the inclusion of trans and non-binary people (e.g., consider over-recruitment, explicitly recruit from trans and non-binary spaces such as Trans Pride parades, hire trans and non-binary people as study staff and recruiters). Although we noted greater diversity among trans and non-binary participants in terms of sexual identities, it is worth noting that this sample was recruited at Pride festivals in urban centres. Accordingly, this study and report do not fully represent the experiences of heterosexual trans men and heterosexual non-binary people nor trans and non-binary people who do not live in or visit urban centres.

In the survey, we asked two questions about sex and gender to delineate the categories of transgender and non-binary for our analysis, as described in the Methods section of this paper. First, "What is your gender identity?" And second, "Do you have trans experience?" The latter question was used as an attempt to be inclusive of those who have transitioned but do not use the term transgender to (currently) describe their identity. A different two-step method of determining who is trans has been utilized in other publications [33]. This includes asking gender identity and sex assigned at birth. Using these questions would make our data more easily comparable with other studies, yet, during consultations with community members, we received feedback that the question of sex assigned at birth was invasive and not necessary for our purposes. Due to our method of gender grouping and the small number of participants who identified as non-binary and not trans (n = 44), we were unable to directly compare trans and non-binary respondents for this analysis as the two categories largely

overlapped; this may also influence our interpretation of results. In the future, analyses that allow for more nuanced comparison across gender categories (e.g., cisgender, exclusively trans, exclusively non-binary, both trans and non-binary) may yield different findings. Additionally, gender groupings for this report were determined through self-reported responses to two survey questions, so participants who skipped these questions or answered "prefer not to say" are not represented in these results.

Future intersectional research may provide valuable insights into the health of trans and non-binary people in Canada [34]. Several areas of inquiry in this study would benefit from more research. First, the ethnoracial make-up of our sample is notable with a large portion of trans and non-binary people identifying as Indigenous; there is little research on ethnicity and race within the trans population. Of note, we did not fully or explicitly address Two-Spirit participants into this analysis; additional resources should be invested into Indigenous Two-Spirit research to produce culturally relevant knowledge on the unique experiences of Indigenous and Two-Spirit people. Our sample did not show statistically significant differences between substance use for the groups we compared, while other studies have found higher rates of use among trans people than cisgender people [35, 36]. Other studies have theorized substance use as a coping mechanism for people who experience marginalization, including trans and non-binary people [37]. It is unclear why our sample does not reflect the findings of others. In the future, qualitative data collection may be useful in understanding this phenomenon. Lastly, a portion of our trans and non-binary respondents indicated being denied access to hormone therapy or gender-affirming surgery; with no follow-up questions, we cannot explain why these individuals were denied the care they asked for. Gender-affirming surgery has been shown to increase mental health and overall well-being for those who need and access it [38, 39]. Those denied surgery may, therefore experience poor mental health, overall health or well-being. Future research should examine the reasons for being denied care and determine ways to support trans and non-binary people's health until they can access these essential services.

## Conclusion

In summary, trans and non-binary people experience significant disadvantages compared with cisgender sexual minority men. We highlight a number of areas for future research and interventions to understand and address health and social inequities of trans and non-binary people with respect to education, employment, mental health, and sexual health. Improved educational supports and employment protections, access to queer and gender affirming healthcare, and trauma-informed mental health services are needed to improve the health and wellbeing of trans and non-binary people in Canada.

## Acknowledgments

The authors would like to thank the 2018 Sex Now Survey participants, the Community-Based Research Centre, and all our community partners for assisting in data collection for this study. Additionally, we would like to thank our trans and non-binary community consultants and members of CBRC's Research Working Group for their input on our preliminary results, which are available in a publicly accessible report: https://www.cbrc.net/community_profiles_trans_non_binary_people.

## Author Contributions

**Conceptualization:** Leo Rutherford, Kiffer G. Card, Nathan J. Lachowsky.

**Data curation:** Aeron Stark.

**Formal analysis:** Aeron Stark, Aidan Ablona, Kiffer G. Card, Nathan J. Lachowsky.

**Funding acquisition:** Nathan J. Lachowsky.

**Investigation:** Robert Higgins, Nathan J. Lachowsky.

**Methodology:** Aidan Ablona, Nathan J. Lachowsky.

**Project administration:** Benjamin J. Klassen, Robert Higgins, Christopher J. Draenos.

**Supervision:** Robert Higgins, Nathan J. Lachowsky.

**Writing – original draft:** Leo Rutherford, Aeron Stark, Benjamin J. Klassen, Hanna Jacobsen, Nathan J. Lachowsky.

**Writing – review & editing:** Leo Rutherford, Benjamin J. Klassen, Nathan J. Lachowsky.

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
