## [Decision Letter · Decision Letter 0]

22 Oct 2020

PONE-D-20-30194

Health and well-being of trans and non-binary participants in a community-based survey of gay, bisexual, and queer men, and non-binary and Two-Spirit people across Canada

PLOS ONE

Dear Dr. Klassen,

Thank you for submitting your manuscript to PLOS ONE. After careful consideration, we feel that it has merit but does not fully meet PLOS ONE’s publication criteria as it currently stands. Therefore, we invite you to submit a revised version of the manuscript that addresses the points raised during the review process.

I sent your manuscript for review to two experts in the field, both of whom agreed there were many positive aspects of the study and the written report. At the same time, reviewers raised several important points that I would invite you to consider before the manuscript can be considered for publication. For all reviewer feedback, please provide a point-by-point response. While many issues raised are more focused on technical changes to the manuscript, some do involve ways of rethinking the analyses and you should carefully consider these and provide a strong response if you elect not to pursue such recommendations.

We look forward to receiving your revised manuscript.

Kind regards,

H. Jonathon Rendina, PhD, MPH

Academic Editor

PLOS ONE

Journal Requirements:

2. Please state in your methods section whether you obtained consent from parents or guardians of the minors included in the study or whether the research ethics committee or IRB approved the lack of parent or guardian consent.

3.We note that you have indicated that data from this study are available upon request. PLOS only allows data to be available upon request if there are legal or ethical restrictions on sharing data publicly. For information on unacceptable data access restrictions, please see http://journals.plos.org/plosone/s/data-availability#loc-unacceptable-data-access-restrictions.

Reviewers' comments:

Reviewer's Responses to Questions

**Comments to the Author**

1. Is the manuscript technically sound, and do the data support the conclusions?

Reviewer #1: Yes

Reviewer #2: Yes

2. Has the statistical analysis been performed appropriately and rigorously? 

Reviewer #1: Yes

Reviewer #2: Yes

3. Have the authors made all data underlying the findings in their manuscript fully available?

Reviewer #1: No

Reviewer #2: No

4. Is the manuscript presented in an intelligible fashion and written in standard English?

Reviewer #1: Yes

Reviewer #2: Yes

5. Review Comments to the Author

Reviewer #1: This is a very informative and well-written paper. I thoroughly enjoyed reading it. Just a few minor suggestions:

1) If possible (word count permitting), could you make it more clear in the abstract that cisgender women were the only potential participants excluded based on gender?

2) In the methods section, could you provide the thresholds/criteria used to determine statistical significance?

3) In the results section, I'm assuming you mean "statistically significant" when you say significant differences?

4) For the adjusted ORs (starting with Table 2) in each health topic area (e.g. mental health, sub use, STIs, etc.), were each of these variables assessed in separate models? Each category has an AOR, but from the methods section it sounds like there were only 5 confounders (so I'm assuming they weren't all in the same model or even topic-based models). So were these 5 confounders used for ALL of these variables of interest? More clarification in the model construction would help.

That's all my comments. Really well-written and well done.

Reviewer #2: General - Terminology:

1. Authors use various terminology to describe sexual minority men. In row 40, I would introduce ‘sexual minority men’ as an umbrella term.

2. Authors use language ‘minoritized’ to describes sexual and gender minority people. Highly recommend the authors modify the language from ‘minoritized’ to ‘minority’, which is consistent with the current literature.

3. I would strongly recommend some additional context for the designation of the ‘non-binary’ group. Authors categorized participants in this group who responded “neither” to the question “What is your gender identity?” with the possible responses being: “man” “woman” or “neither. I prefer to self-describe as: _____________.”

From my reading of the article, authors should define the term ‘non-binary’ and its rationale for use based on the response options provided above. Since ‘non-binary’ is an identity label, it is very possible that participants self-described their identity as non-binary in the open text field. But for those participants who didn’t, it needs to be clear that its use in this article is more of an umbrella term.

Abstract:

Row 32: Please add comparison group (cisgender sexual minority men)

Introduction:

Row 49: Authors provided an example of a healthcare disparity – I would suggest changing the term ‘health disparities’ to ‘healthcare disparities’.

Row 64: Health disparities are introduced here. I would recommend modifying this sentence: “Additionally, trans and non-binary people experience health disparities, and report poorer physical and mental health than their cisgender peers”.

Row 84: Recommend changing “statistics” to “findings”.

Row 84: It is unclear if these findings are associated with the Trans PULSE study or the qualitative interviews conducted by Rich et al.: “Some participants were living in an urban centre and well-connected to healthcare systems, such as routine transition-related care appointments that made integrating HIV testing into their regular healthcare regimen easy”. Please clarify.

Row 98: I would recommend clarifying the time-period for ‘did not have enough to eat’

Row 155: I would recommend moving the social media recruitment efforts earlier in the paragraph with the other recruitment efforts.

Results

Due to the questions about gender identity and trans experience, the authors indicated they are unable to provide statistical comparisons between the those who identified as trans and those who were categorized in the non-binary group and acknowledge this as a limitation. I would suggest that the authors discuss rationale for not using an analysis plan with four categories: (1) cisgender, (2) trans, (3) non-binary, and (4) non-binary with trans experience.

6. PLOS authors have the option to publish the peer review history of their article (what does this mean?). If published, this will include your full peer review and any attached files.

Reviewer #1: No

Reviewer #2: No

---

## [Author Response · Author response to Decision Letter 0]

9 Dec 2020

(See attached Response to Reviewers document for formatted version)

RESPONSE TO REVIEWERS 

Thank you for the opportunity to revise and resubmit this manuscript for consideration in PLOS ONE. We are appreciative of the comments provided by the editor and reviewers, which have strengthened this manuscript. Authors responses are in bullet-points and bolded text below reviewer comments.

We would like to amend our Financial Disclosure statement to the following:

Sex Now 2018 received funding support from Canadian Blood Services MSM Research Fund, funded by the federal government (Health Canada) and the provincial and territorial ministries of health. The views herein do not necessarily reflect the views of Canadian Blood Services or the federal, provincial, or territorial governments of Canada. Additional in-kind contributions were received from the Public Health Agency of Canada's National Microbiology Laboratory. Additional funding for this analysis was provided by Women and Gender Equality (WAGE) Canada. NJL is supported by a Scholar Award from the Michael Smith Foundation for Health Research (#16863).

When submitting your revision, we need you to address these additional requirements. Please ensure that your manuscript meets PLOS ONE's style requirements, including those for file naming. The PLOS ONE style templates can be found at https://journals.plos.org/plosone/s/file?id=wjVg/PLOSOne_formatting_sample_main_body.pdf and https://journals.plos.org/plosone/s/file?id=ba62/PLOSOne_formatting_sample_title_authors_affiliations.pdf

• We have made changes to the manuscript based on the provided submission guidelines.

Please state in your methods section whether you obtained consent from parents or guardians of the minors included in the study or whether the research ethics committee or IRB approved the lack of parent or guardian consent.

• Thank you. We have clarified that no parental consent was required per our REB approval in lines 180 and 181 of the clean manuscript.

We note that you have indicated that data from this study are available upon request. PLOS only allows data to be available upon request if there are legal or ethical restrictions on sharing data publicly. For information on unacceptable data access restrictions, please see http://journals.plos.org/plosone/s/data-availability#loc-unacceptable-data-access-restrictions. In your revised cover letter, please address the following prompts: a) If there are ethical or legal restrictions on sharing a de-identified data set, please explain them in detail (e.g., data contain potentially identifying or sensitive patient information) and who has imposed them (e.g., an ethics committee). Please also provide contact information for a data access committee, ethics committee, or other institutional body to which data requests may be sent.

If there are no restrictions, please upload the minimal anonymized data set necessary to replicate your study findings as either Supporting Information files or to a stable, public repository and provide us with the relevant URLs, DOIs, or accession numbers. 

Please see http://www.bmj.com/content/340/bmj.c181.long for guidelines on how to de-identify and prepare clinical data for publication. For a list of acceptable repositories, please see http://journals.plos.org/plosone/s/data-availability#locrecommended-repositories."

• The University of Victoria's Human Research Ethics Board has only approved storage of our data on secure university servers since the data contain potentially sensitive information about study participants. Data is available on request through secure university servers only. Any requests to access the data can be made to Dr. Nathan Lachowsky at the University of Victoria (nlachowsky@uvic.ca).

Reviewers' comments: Reviewer's Responses to Questions Comments to the Author 1.

Is the manuscript technically sound, and do the data support the conclusions? The manuscript must describe a technically sound piece of scientific research with data that supports the conclusions. Experiments must have been conducted rigorously, with appropriate controls, replication, and sample sizes. The conclusions must be drawn appropriately based on the data presented. Reviewer #1: Yes Reviewer #2: Yes 

• Response Not Applicable 

Has the statistical analysis been performed appropriately and rigorously? Reviewer #1: Yes Reviewer #2: Yes

• Response Not Applicable 

Have the authors made all data underlying the findings in their manuscript fully available? The PLOS Data policy requires authors to make all data underlying the findings described in their manuscript fully available without restriction, with rare exception (please refer to the Data Availability Statement in the manuscript PDF file). The data should be provided as part of the manuscript or its supporting information, or deposited to a public repository. For example, in addition to summary statistics, the data points behind means, medians and variance measures should be available. If there are restrictions on publicly sharing data—e.g. participant privacy or use of data from a third party—those must be specified. Reviewer #1: No Reviewer #2: No

• The University of Victoria's Human Research Ethics Board has only approved storage of our data on secure university servers since the data contain potentially sensitive information about study participants. Data is available on request through secure university servers only. Any requests to access the data can be made to Dr. Nathan Lachowsky at the University of Victoria (nlachowsky@uvic.ca).

Is the manuscript presented in an intelligible fashion and written in standard English? PLOS ONE does not copyedit accepted manuscripts, so the language in submitted articles must be clear, correct, and unambiguous. Any typographical or grammatical errors should be corrected at revision, so please note any specific errors here. Reviewer #1: Yes Reviewer #2: Yes

• Response Not Applicable 

Review Comments to the Author Please use the space provided to explain your answers to the questions above. You may also include additional comments for the author, including concerns about dual publication, research ethics, or publication ethics. (Please upload your review as an attachment if it exceeds 20,000 characters)

If possible (word count permitting), could you make it more clear in the abstract that cisgender women were the only potential participants excluded based on gender?

• Line 27 includes a sentence clarifying that all people who identified as women were ineligible

 In the methods section, could you provide the thresholds/criteria used to determine statistical significance?

• Thank you. We have added text in the methods section to provide this information in lines 214-215 of the clean manuscript.

 In the results section, I'm assuming you mean "statistically significant" when you say significant differences?

• Correct. We have added qualifiers in throughout results to clarify we mean “statistically significant.” 

 For the adjusted ORs (starting with Table 2) in each health topic area (e.g. mental health, sub use, STIs, etc.), were each of these variables assessed in separate models? Each category has an AOR, but from the methods section it sounds like there were only 5 confounders (so I'm assuming they weren't all in the same model or even topic-based models). So were these 5 confounders used for ALL of these variables of interest? More clarification in the model construction would help. That's all my comments. Really well-written and well done.

• We have added content in lines 212-213 and 290-291 to clarify that each variable was assessed in separate models, and adjusted for the same 5 confounders.

General - Terminology:

Authors use various terminology to describe sexual minority men. In row 40, I would introduce ‘sexual minority men’ as an umbrella term.

• Thank you. We have added the phrase ‘sexual minority’ men and replaced instances of ‘minoritized’ with ‘minority’. 

Authors use language ‘minoritized’ to describes sexual and gender minority people. Highly recommend the authors modify the language from ‘minoritized’ to ‘minority’, which is consistent with the current literature.

• See comment directly above.

I would strongly recommend some additional context for the designation of the ‘non-binary’ group. Authors categorized participants in this group who responded “neither” to the question “What is your gender identity?” with the possible responses being: “man” “woman” or “neither. I prefer to self-describe as: _____________.” From my reading of the article, authors should define the term ‘non-binary’ and its rationale for use based on the response options provided above. Since ‘non-binary’ is an identity label, it is very possible that participants self-described their identity as non-binary in the open text field. But for those participants who didn’t, it needs to be clear that its use in this article is more of an umbrella term.

• We have added content in rows 197-200 to give context and rationale for the non-binary group.

Abstract: Row 32: Please add comparison group (cisgender sexual minority men) 

• We have added the comparison group.

Introduction: Row 49: Authors provided an example of a healthcare disparity – I would suggest changing the term ‘health disparities’ to ‘healthcare disparities’.

• We have changed the term to healthcare disparities.

Row 64: Health disparities are introduced here. I would recommend modifying this sentence: “Additionally, trans and non- binary people experience health disparities, and report poorer physical and mental health than their cisgender peers”.

• We have edited this sentence as suggested.

Row 84: Recommend changing “statistics” to “findings”. Row 84: It is unclear if these findings are associated with the Trans PULSE study or the qualitative interviews conducted by Rich et al.: “Some participants were living in an urban centre and well-connected to healthcare systems, such as routine transition-related care appointments that made integrating HIV testing into their regular healthcare regimen easy”. Please clarify.

• We have changed ‘statistics’ to ‘findings’ and clarified that findings referenced the Rich et al study.

Row 98: I would recommend clarifying the time-period for ‘did not have enough to eat’

• We have clarified that this was asked for a 12-month period. 

Row 155: I would recommend moving the social media recruitment efforts earlier in the paragraph with the other recruitment efforts.

• We have moved this earlier in the paragraph.

Results: Due to the questions about gender identity and trans experience, the authors indicated they are unable to provide statistical comparisons between the those who identified as trans and those who were categorized in the non-binary group and acknowledge this as a limitation. I would suggest that the authors discuss rationale for not using an analysis plan with four categories: (1) cisgender, (2) trans, (3) non-binary, and (4) non-binary with trans experience.

• We have added clarification and rationale for our gender groupings in the results section (lines 577-582 of the clean manuscript).

---

## [Decision Letter · Decision Letter 1]

21 Jan 2021

Health and well-being of trans and non-binary participants in a community-based survey of gay, bisexual, and queer men, and non-binary and Two-Spirit people across Canada

PONE-D-20-30194R1

Dear Dr. Klassen,

We’re pleased to inform you that your manuscript has been judged scientifically suitable for publication and will be formally accepted for publication once it meets all outstanding technical requirements.

Kind regards,

H. Jonathon Rendina, PhD, MPH

Academic Editor

PLOS ONE

Additional Editor Comments (optional):

Reviewers' comments:

Reviewer's Responses to Questions

**Comments to the Author**

1. If the authors have adequately addressed your comments raised in a previous round of review and you feel that this manuscript is now acceptable for publication, you may indicate that here to bypass the “Comments to the Author” section, enter your conflict of interest statement in the “Confidential to Editor” section, and submit your "Accept" recommendation.

Reviewer #2: All comments have been addressed

2. Is the manuscript technically sound, and do the data support the conclusions?

Reviewer #2: Yes

3. Has the statistical analysis been performed appropriately and rigorously? 

Reviewer #2: Yes

4. Have the authors made all data underlying the findings in their manuscript fully available?

Reviewer #2: No

5. Is the manuscript presented in an intelligible fashion and written in standard English?

Reviewer #2: Yes

6. Review Comments to the Author

Reviewer #2: The authors have thoroughly addressed all comments and have provided a rationale for why the data that underlies the findings described in their manuscript are not fully available without restriction.

7. PLOS authors have the option to publish the peer review history of their article (what does this mean?). If published, this will include your full peer review and any attached files.

Reviewer #2: No

---

## [Editor Report · Acceptance letter]

26 Jan 2021

PONE-D-20-30194R1 

Health and well-being of trans and non-binary participants in a community-based survey of gay, bisexual, and queer men, and non-binary and Two-Spirit people across Canada 

Dear Dr. Klassen:

I'm pleased to inform you that your manuscript has been deemed suitable for publication in PLOS ONE. Congratulations! Your manuscript is now with our production department. 

Kind regards, 

on behalf of

Dr. H. Jonathon Rendina 

Academic Editor

PLOS ONE